# Physico-Chemical Characterization of Keratin from Wool and Chicken Feathers Extracted Using Refined Chemical Methods

**DOI:** 10.3390/polym15010181

**Published:** 2022-12-30

**Authors:** Sara Mattiello, Alessandro Guzzini, Alessandra Del Giudice, Carlo Santulli, Marco Antonini, Giulio Lupidi, Roberto Gunnella

**Affiliations:** 1Physics Section, School of Science and Technology, Università di Camerino, via Madonna delle Carceri, 62032 Camerino, Italy; 2School of Bioscience and Veterinary Medicine, Università di Camerino, via Gentile III da Varano, 62032 Camerino, Italy; 3Department of Chemistry, Sapienza Università di Roma, Piazzale Aldo Moro 5, 00185 Rome, Italy; 4Geology Section, School of Science and Technology, Università di Camerino, via Gentile III da Varano 7, 62032 Camerino, Italy; 5ENEA—SSPT BIOAG PROBIO Via Gentile III da Varano, 62032 Camerino, Italy

**Keywords:** keratin, chicken feather waste, wool waste, metabisulfite extraction, Raman, SAXS

## Abstract

In this work, the characteristic structure of keratin extracted from two different kinds of industrial waste, namely sheep wool and chicken feathers, using the sulfitolysis method to allow film deposition, has been investigated. The structural and microscopic properties have been studied by means of scanning electron microscopy (SEM), Raman spectroscopy, atomic force microscopy (AFM), and infrared (IR) spectroscopy. Following this, small-angle X-ray scattering (SAXS) analysis for intermediate filaments has been performed. The results indicate that the assembly character of the fiber can be obtained by using the most suitable extraction method, to respond to hydration, thermal, and redox agents. The amorphous part of the fiber and medium range structure is variously affected by the competition between polar bonds (reversible hydrogen bonds) and disulfide bonds (DB), the covalent irreversible ones, and has been investigated by using fine structural methods such as Raman and SAXS, which have depicted in detail the intermediate filaments of keratin from the two different animal origins. The preservation of the secondary structure of the protein obtained does offer a potential for further application of the waste-obtained keratin in polymer films and, possibly, biocomposites.

## 1. Introduction

Keratin represents the most abundant structural protein in epithelial cells, and alongside with collagen, the most important biopolymer in animals. Its use in nature has an influence on its crystallinity and geometrical arrangement. This is connected in turn to its function, whether it serves, e.g., as thermal insulation and moisture control, such as in wool, or it absolves to more structural duties, such as in chicken feathers [1]. Keratin can be self-assembled to be employed for protective purposes, either explicitly as the main constituent of armors, or more implicitly in damage-tolerant structures, providing a shielding effect to the whole animal or to some critical parts of its body [2].

As a matter of fact, though, the function of protection is declined in different ways, which directly derive from the micro-assembly process developed in nature. For example, the cortical cells of animal hair, such as in wool keratin, constitute a microfibrillar composite, obtained with closely packed α-helical low-sulfur subunits [3]. In contrast, feather keratins, with their typical barbs–barbules network, also contain β-pleated sheets and are capable of forming filaments [4].

Large amounts of waste keratin are available from activities such as butchery and animal rearing. In particular, the poultry and wool industries generate huge amounts of keratin waste, estimated worldwide at 8.5 billion tons annually for the former (2018 data) [5] and 5 million tons produced yearly in the EU for the latter in 2019 [6]. The effective and profitable use of this waste would both reduce the amount of harmful material in the environment and the consumption of resources. Improving the extraction method for keratin from wool and feather waste can be a measure inspired by circular economy to reduce the environmental impact subsequent to the disposal of this waste, while providing indications for the “good practice” of its use as a secondary raw material. Landfilling, burying, and incinerating, which are still common procedures for this waste, represent environmental threats also in view of the limited significance, if any, of keratin structures for soil nutrition, due to the limited variety of micro-organisms able to feed on them [7].

Keratinous materials, formed by explicitly organized keratinized cells filled with mainly fibrous proteins (keratins), are natural polymer composites that exhibit a complex hierarchical structure ranging from nanoscale to centimeter scale (Figure 1). More specifically, the definition of “keratins” refers to a group of insoluble proteins and form the bulk of the stratum corneum of the epidermis and the epidermal appendages, such as hair, nails, horns, and feathers [8]. The basic macromolecules that form keratin are polypeptide chains constituted by amino acids. Keratin has a large amount of cysteine residues, which have a thiol group (-SH), producing a strong, covalent disulfide bond that cross-links both the polypeptide chains and the matrix molecules together [9]. Wool and feathers are characterized by two kinds of filament matrix structures at the nanoscale: alpha and beta keratin.

In particular, the alpha keratin proteins are organized as spiral coils. The filament twists along its length in a right-handed coil; two chains formed by disulfide cross-link a left-handed coil, named as the dimer (45 nm long). Following this, dimers aggregate end-to-end and stagger side-by-side via disulfide bonds to form a protofilament (approximately 2 nm diameter); two protofilaments laterally associate into a protofibril; four protofibrils combine into a circular or helical intermediate filament with a diameter of 7 nm [10]. In contrast, for beta keratin, the pleated sheet consists of laterally packed strands which can be parallel or antiparallel, forming chains that are held together by intermolecular hydrogen bonds. The pleated sheet structure is stabilized by two factors: the hydrogen bonds between beta strands contribute to form a sheet, while the peptide bond forces a beta-sheet to be pleated. The beta keratin filament is formed from the folded sheet that twists into a left-handed helix. Two pleated sheets overlap and wind in opposite directions, forming a filament with a diameter of 4 nm [11].

Keratins can be classified as soft keratin and hard keratin. The difference between the two kinds of keratin is the quantity of sulfur and the ability that proteins have to be bonded. Soft keratins, typically present in stratum corneum, have a lower amount of sulfur and are weakly consolidated, whilst hard keratins, found in hair, nails, and feathers, have a more coherent structure and a higher amount of sulfur [12].

The aim of the present work is to improve the knowledge of the process of extraction of the raw materials from wool and feathers. The complete chemical–physical analysis of the two different extracted keratins provides useful structural information for the definition of the future applications of this type of waste. These would possibly range from the introduction of keratin from different kinds of waste as filler in biodegradable films, to the production of films using extracted keratin.

## 2. Materials and Methods

### 2.1. Waste Materials

Keratin waste materials were obtained from chicken slaughtering and sheep shearing, without any treatment. The wool analyzed comes from flocks of merino and merinized sheep of the Apennine ridge, across the regions of Marche, Umbria, and Abruzzo (Italy). A sample of as-received waste is portrayed in Figure 2.

The two types of as-received waste are widely recognized as containing a very high amount of pure keratin. In particular, as concerns chicken feathers, these are essentially constituted (>90%) by a structural keratin structure, rich in cysteine, and hydrophobic residues that enhance cross-linking by disulfide bonds, including a variety of other amino acids, such as lysine, serine, and proline [13,14]. On the other hand, wool was proven to contain up to 95% keratin by weight and all nine essential amino acids, so as to be considered a pure source of intermediate filament proteins [15]. The aforementioned evidence justified the selection of both sources for the extraction of keratin.

### 2.2. Extraction Method

A variety of chemical methods are potentially available for the extraction of keratin from industrial waste. For example, in the case of sheep wool, methods such as alkali hydrolysis, sulfitolysis, reduction, oxidation, and extraction using ionic liquid, were considered [16]. It is important to clarify that with sulfitolysis, reduction, and oxidation, only the cleavage of disulfide bonds takes place, whereas keratin hydrolysis leads to peptide bonds’ breakage, which may be undesirable in view of the possible upcycling of waste in mechanically sound structures e.g., films [17]. However, it has been highlighted that, other than the yield of extraction, antioxidant properties of the extracted keratin from chicken poultry may vary e.g., using 2-mercaptoethanol, sodium sulfite, and sodium dodecyl sulfate [18]. The metabisulfite extraction method was selected as it provides sufficient yield, whilst also preserving the secondary structure of the protein [19]. In addition, the procedure is effective, being easily applicable to the raw material and requires lower amounts of chemicals and with lower toxicity compared to other methods such as the mercaptoethanol one [20,21]. The extraction of keratin from wool and chicken feathers involved a first wash with cold water and soap, followed by drying to constant weight at 60 °C in a forced-air oven. Dried sheep wool and chicken feathers were ground to pieces with maximal dimension not exceeding 5 mm. The dried wool and feathers were incubated with ethanol at 50 °C for 2 h to remove surface fats and waxes, which were then filtered out. All residues of ethanol on defatted wool or feathers were removed after incubation for 3 h in a forced-air oven at 60 °C. Subsequently, 3 4g of clean and defatted wool were added to 100 mL of a solution containing 48 g of urea (8 M), 1.8 g of sodium dodecyl sulfate (SDS), and 6.84 g of sodium metabisulphite. The reaction took place at 70 °C for 24 h under shaking, and after incubation, filtered for solids removal (partially digested residue). The filtered solution was dialyzed against ultrapure water for approximately 3 days at room temperature with several water changes using 12 kDa nominal low-molecular-weight cutoff membrane. After dialysis, the solution was centrifuged to separate the precipitated protein and the supernatant obtained was lyophilized to obtain a pure keratin powder. The process to obtain keratin from cleaned and defatted feathers was carried out with the following process: 4 g of clean feathers were immersed in 150 mL of a solution containing 47.7 g of urea, 7.5 g of SDS, 15 g of sodium metabisulfite, 195 mg of ethylenediaminetetraacetic acid (EDTA), and 4.84 g of tris(hydroxymethyl) aminomethane (Tris). The reaction took place at 70 °C for 24 h under shaking, as described above for wool. This method of extraction applied to wool and feathers allowed us to obtain keratin in powder form.

To determine the purity and an average molecular mass of the keratin hydrolysate samples, SDS-PAGE gel electrophoresis was performed according to the method of Laemmli [22]. The samples of freeze-dried lyophilized hydrolysate were mixed (at a ratio of 4:1) with a loading buffer 5x (containing 10% (*v*/*v*) SDS, 250 mM Tris−HCl buffer (pH 6.8), 50% (*v*/*v*) glycerol, 0.5 M dithiothreitol (DTT), 0.02% (*w*/*v*) bromophenol blue) and the solutions were heated in a dry bath heat block at 95 °C for 5 min. Denatured keratin samples from feather (F) and wool (W) were loaded onto each lane, resolved on a precast polyacrylamide separation gel 4–20%, and stained with 0.25% (*w*/*v*) Coomassie brilliant blue R250. Denatured protein markers, having a known molecular weight, were used as a standard. The different lyophilized keratins used in the experiments were not further purified by size exclusion chromatography.

### 2.3. Characterization of the Extracted Keratin

#### 2.3.1. Keratin Film Preparation

Films were prepared by drop casting, to be subsequently characterized by scanning electron microscope (SEM) and atomic force microscope (AFM). For the deposition, 10 mg of keratin was dissolved in 1 mL of de-ionized water. Given the low solubility of keratin in water, dissolution was facilitated by using the sonicator for 10 min until a uniform suspension was obtained without precipitation for several days. A droplet of approximately 10 µL of the solution was deposited on glass forming a thin layer of keratin with a nominal thickness of approximately 4.8 µm, as estimated from the density of the material and the subsequent volume obtained from measuring the film weight. This value is, of course, only approximated and is measured considering the density as constant throughout the keratin layer. The film was dried under fume hood at room temperature.

#### 2.3.2. Microscopy Techniques

SEM analyses were carried out using a Field Emission Scanning Electron Microscopy (FE-SEM, Sigma Family, Zeiss, Jena, Germany), equipped with a backscattered detector (BSD) to obtain high-quality microphotographs.

AFM images were acquired in tapping mode by using CSI Nano-observer, Les Ulis, France, and P-doped n-type Si cantilever (resonance frequency = 75 kHz). The measurements were performed by using resonant mode.

#### 2.3.3. Spectroscopy Techniques

Raman spectroscopy and FT-IR were used to investigate the secondary structure of keratin on powder samples.

Raman spectroscopy was performed with a HORIBA IHR320 micro-Raman Scattering system (Horiba, Palaiseau, France) equipped with an optical Microscope model Olympus BXF41 (with 5×, 20×, 50×, 100× objectives) (Münster, Germany). The Raman spectrometer was operated at 532 nm (diode laser).

IR spectra were recorded from 4000 to 600 cm^−1^ with a PerkinElmer Spectrum 100 FT-IR instrument (Waltham, MA, USA) by total reflectance on a CdSe crystal.

#### 2.3.4. Small-Angle X-ray Scattering (SAXS)

Small-angle X-ray scattering experiments were performed using a Xeuss 2.0 Q Xoom system (Xenocs SA, Grenoble, France) equipped with a micro-focus Genix 3D X-ray Cu source (λ = 0.1542 nm), a two-dimensional Pilatus3 R 300 K detector placed at variable distance from the sample (Dectris Ltd., Baden, Switzerland).

Measurements were made on liquid solutions of keratin extracted from wool and feathers at different concentrations (0.5, 0.7, 1, 1.2 wt.%) obtained by dissolving weighted amounts of keratin powder with 1 mL of Tris-HCl 50 mM buffer (pH 8).

The keratin samples were loaded into disposable glass capillary cells with nominal thickness 1.5 mm and sealed with hot glue. Two capillaries, one loaded with the buffer used as dispersant and an empty one, were used for background subtraction.

The measurements were performed at room temperature (25 ± 1 °C) and at reduced pressure (∼0.2 mbar), with two different sample-detector distances, in order to overall access a scattering vector modulus (q) range between 0.045 and 13 nm^−1^, where q = 4πsin(θ)/λ, and 2θ is the scattering angle. The two-dimensional scattering patterns were subtracted for the “dark” counts, and then masked, azimuthally averaged, and normalized for transmitted beam intensity, exposure time, and subtended solid angle per pixel, by using the FoxTrot software developed at SOLEIL. The one-dimensional intensity vs. q profiles were subtracted for the contributions of the solvent and empty capillary and measured in intensity units of macroscopic scattering cross-section (cm^−1^) by dividing them for the capillary thickness estimated from the alignment scans. Pair distance distributions of the cross-section of elongated objects were obtained by indirect Fourier inversion of the I(q)·q profiles performed with the software BayesApp [23].

## 3. Results and Discussion

The sodium dodecyl sulfate polyacrylamide gel electrophoresis (SDS-PAGE) of keratins’ preparations is reported in Figure 3.

The molecular weights of extracted protein fractions were in the range of 12–13 kDa for feathers (F), in the range of what was observed in [24], and the protein extracted seems to be relatively pure, near homogeneity. On the other hand, for wool (W) a large electrophoretic band with a similar MW to that reported for feathers is reported, in addition with another less intense band around 18 kD.

The yield of each extraction method (Y) was calculated in percent by the ratio between the weight of the freeze-dried sample (W′) and initial weight of the sample (W) using the following Equation (1):Y (%) = (W′/W) × 100(1)

The yield of the extraction method, measured over three extractions for each waste, was 45 (±3)% for wool, whereas the yield obtained with feathers was 73 (±3)%. The data obtained for wool, although they might appear quite low, were even slightly higher than what was reported from sulfitolysis elsewhere, i.e., 41% [17]. The remainder is a partially digested residue that can be dried in the oven and mechanically reduced to powder.

Images of the extracted portion and the relevant residues are reported in Figure 4.

Optical micrographs of the residues of the keratin extraction are reported in Figure 5. Here, the needle-like structure with random orientation and very variable aspect ratios, with length mostly in the range between 10 and 20 microns, obtained from feather waste (Figure 5a) and the filamentous structure obtained from wool waste (Figure 5b), are respectively shown.

Moreover, the microscopic characters of the films were analyzed with atomic force microscope and scanning electron microscope. Atomic force microscope images of keratin extracted from wool (Figure 6a) show the topography of the film at different dimensions. The structures are poorly defined and globular with a radius around 45 nm. In contrast, the images of keratin extracted from feathers (Figure 6b) show the topography of the film with elongated structures that have lengths of approximately 500 nm. This is confirmed by SEM images (Figure 7), where the image of the keratin extracted from wool (Figure 7a) is globular and compact and does not show elongated structures. Otherwise, the keratin extracted from feathers (Figure 7b) has a uniform surface with elongated structures with a section of approximately 0.5 µm. No evidence of preferential orientation is clearly observable.

Raman spectroscopy was also used to investigate the secondary structure of keratin on powder samples. Figure 8 shows spectra of keratin extracted from wool (in black) and feathers (in red) and the position of the main bands are 1200–1300 cm^−1^ for amide III band, 1600–1700 cm^−1^ for amide I band, and 1448 cm^−1^ for CH_2_ group [25].

The spectra confirm that after the extraction, the keratin preserves its secondary protein structure, and it is possible to obtain further details about the structure by studying in more detail the different contributions that compose the amide I band. In particular, the position of two evident peaks have been identified at 1609 cm^−1^ for C = C double bond and between 1651 cm^−1^ and 1679 cm^−1^ for alpha and beta-sheet. Signals coming from the two latter contributions are partially superposed, yet a deconvolution operation (not shown) highlights the predominance of an alpha-helix structure (1651 cm^−1^) in the case wool keratin and beta-sheet structure (1679 cm^−1^) in the case of feather keratin.

The IR spectra of keratin powder extracted from wool and feathers are shown in Figure 9. Vibrations due to the characteristic bands of proteins are visible in both spectra. The absorption band of stretching vibration of N-H and OH bonds at around 3300 cm^−1^ is related to amide A [26]. Stretching vibrations of C=O bonds appear from 1600 to 1700 cm^−1^ and they are typical of the amide I band, that contains information about the secondary structure of keratin [27]. At 1520 cm^−1^, the bending vibration of N-H of amide II is visible [28]. The stretching vibrations of C-N and C–H and bending vibrations of N-H and C=O at around 1220–1300 cm^−1^ are related to amide III. Additionally, in this case, it is possible to obtain further information about the secondary structure by studying the contribution that constitutes the amide I band. In particular, the position of the amide I peak for wool keratin has been identified deconvoluted in one component at 1618 cm^−1^ for beta-sheet, one at 1645 cm^−1^ for alpha-helix, and one at 1675 cm^−1^ for disordered structure, while for feather keratin the peaks are in position 1625 cm^−1^ for beta-sheet, 1650 cm^−1^ for alpha-helix, and 1676 cm^−1^ for disordered structure.

Since there is no water interference in the amide III region (1220–1300 cm^−1^), it is possible to use this region to determine protein secondary structure, especially in the presence of pure proteins with a well-defined secondary structure, such as it is the case of feather keratin sample (beta-sheet structure) and wool (alpha keratin). As a matter of fact, the maximum absorbance wavenumber of pure protein containing only alpha-helix is around 1300 cm^−1^, for beta-sheet proteins it is around 1235 cm^−1^, beta-turn bands are located around 1260–1280 cm^−1^, while the random coil is located around 1240–1260 cm^−1^ [29]. However, the presence of a non-homogeneous type of peptides (with different molecular weight derived from the type of extraction conditions) and the presence of oxidized sulfur derivatives such as sulfonic (RSO_3_H) or sulfinic acid (RSO_2_H) showed a more complicated pattern [30]. This could be further investigated in presence of homogeneous solution of the two keratins using the amide III region.

Furthermore, the area between 700 and 1100 cm^−1^ is strongly sensitive to the presence of sulfur-oxidized derivatives, which suggests an increasing number of disulfide bonds has been reduced to form cysteic acid [27]. As regards other peaks, in particular the one detected at 625 cm^−1^, it is probably due to the vibration of the C-S bonds [31]. In contrast, there appears to be no interference due to SDS, even if it has bands in common with keratin: SDS shows intense peaks at 827 cm^−1^ and 1467 cm^−1^, not visible in the spectrum of keratin extracted from feathers, yet due to the aliphatic CH_2_ vibrations of SDS [32].

IR spectroscopy confirms what was detected with Raman spectroscopy, i.e., a greater presence of alpha-helix in the sample of keratin extracted from wool and beta-sheet in the keratin extracted from feathers. It also shows a contribution of disordered structure probably due to the extraction method. These unfolded protein parts are present in both samples.

The SAXS study was aimed at the characterization of the aggregation state of individual keratin filaments (namely intermediate filaments, with expected diameter of 7 nm [9]) when extracted from the raw materials by a sulfitolysis chemical digestion process and redispersed in aqueous solvent. Rather than describe the interaction of such filaments among themselves within a formed film, at this stage the structure of isolated filaments with a diameter in the range between 1 nm and 100 nm was characterized by means of SAXS, and information on the internal structure of the fiber was obtained [33].

For this reason, the choice was to study rather diluted samples (of the order of 10 mg/mL) to reduce the inter-particle interactions. The profiles collected for the two forms of keratin (Figure 10) show an initial power law in the low-q region (q < 0.15 nm^−1^) close to q^−1^, suggesting them to fall within the case of the rod-like particle model, for which the overall length lies above the size limit accessible within the available q-range (>100 nm): as the consequence, only information about their cross-section can be obtained. This was conducted by applying the indirect Fourier transform to the scattering intensity multiplied by q to obtain the pair distance distributions of the cross-section (P_CS_(r), inset of Figure 10). The skewed shape of these functions having intense peaks for small distances (<7 nm) and a slowly decaying profile at larger distances suggests that the average section of the fibrils is constituted by the lateral association of fundamental units with a smaller diameter.

In particular, maxima at approximately 1 nm and 3.5 nm could correspond to fundamental building blocks with diameters of 2 nm and 7 nm, potentially compatible with the size of a proto-filament and a proto-fibril up to the full intermediate filament. In the case of feather keratin, the first peak of the distribution function at approximately 1 nm is more pronounced and well resolved, confirming that keratin filaments in feathers have a higher degree of internal order than in the wool at this short range. From the value of r at which the P*_CS_*(r) fall to zero, an estimate of the maximum diameter of the section of the filaments present in solution is obtained: for the keratin extracted from wool, most of the filaments have diameters ≤7 nm (of the same order of an intermediate filament), and only a small fraction contribute with larger diameters; for the keratin extracted from feathers, more filaments with larger diameters are found, with a maximum size up to 20 nm, suggesting a higher degree of lateral association among intermediate filaments in the dispersed aggregates. Further studies on the *modeling* of these fibers are ongoing, involving core–shell cylinder form factors which also include the contribution of unfolded peptide chains in the shell, as suggested by the final slope observed in the SAXS data, following a power law closer to that expected for a coil (q^−2^) rather than that predicted for a sharp interface according to the Porod law (q^−4^) [34].

In general terms, the impact on the ambient of waste from feathers and wool derived from the important food supply chain poses the question of a possible engineered use of such materials for novel applications. It is widely recognized that keratin-based materials offered distinct properties, in particular coupling biodegradability, biocompatibility, and mechanical durability [35]. In addition, keratin-based biomaterials have an intrinsic ability to support cellular proliferation and can be used as sponges, films, and hydrogels for various biomedical applications. These are all possibilities that, to be explored, require a complete characterization of keratin waste, such as that performed on human hair keratin in [36]. Those applications should combine the well-known basic properties of the raw material into composites having the merit to offer sufficient mechanical and interface performance, needed for its industrial process. To this aim, the detailed knowledge of the material structure and its conversion, adaption, and transformation is necessary. The wool and feather keratins can be derived into water-soluble proteins with high molecular weight by means of simple chemical treatments and can be turned into films with plastic properties, which, in the past, was obtained from feathers, albeit by thermal treatments [37].

In the future, to make keratin the mainstream material for various bio-applications, the properties and structures of keratin must be investigated extensively at both the nanoscale and macroscale levels, as well as under chemical conditions for their dissolution and extraction, considering the filament–matrix structure for alpha and beta keratins of wool and feathers, respectively [38].

In this first step, the study on novel extraction methods described above outlined the properties of the pure fibers. In practice, the extraction of keratin from two different animal waste sources, namely sheep wool and chicken feathers, by the sulfitolysis method indicated some differences in extraction yield, in favor of feathers. However, in both cases, the extraction offered considerably higher yield than reported in the literature for other methods [39]. Metabisulfite is also preferable as a solvent due to its non-toxicity, which has been recognized to reduce the contamination effects, e.g., in cereals [40]. It is noteworthy, though, that both protein waste sources offered the possibility to obtain sufficiently sound and repeatable films from stir casting.

It is suggested, from both optical spectroscopies and X-ray scattering, that keratin from feathers would present a more oriented structure, evidenced by the beta-sheets contribution with respect to disordered coils (alpha-helices) and from the more intense small-angle scattering of x-rays, because of their more compact structure. It is suggested that the application of sulfitolysis did contribute to the preservation of compactness, which has not been accounted for when extracting keratin from poultry feathers using other methods, such as ionic liquids (sodium sulfate) [41]. In particular, the main absorption peaks are preserved in FTIR spectroscopy from both sources of keratin with no frequency shifts; therefore, no visible interaction has been detected with the extraction medium, which has been not the case in works using other methods [42]. As regards more specifically SAXS spectra, apart from the aforementioned higher degree of lateral aggregation for feather keratin, no significant changes in the broadness of peaks are observed between the two sources. This is promising for the production of composite films with other polymers (e.g., cellulose), where the structural variability these changes may imply do represent an issue [43]. As the result, a single model has been hypothesized as sufficiently representative for both keratins.

Though an exact model for the scattering object for the latter case is still unclear, such a more compact structure would possibly oppose higher tearing loads, though film plasticization. The choice of plasticizer does represent a crucial stage to improve these properties [44], which is also related to the detailed object description and its chemical functionalization. The likely presence of unfolded proteins in keratin from feathers can be correlated with the higher viscoelastic properties of hydrogels they are able to form with respect to those synthesized using wool, as observed in [45]. As a matter of fact, in the literature, the production of blended films with wool–plasticizer combinations are less diffuse than with feather keratin, using typical plasticizers such as citric acid [46] or glycerol [47], where the lower properties might be compensated by a higher film translucency. In any case, these studies seldom posed the question of reducing the residue by optimizing keratin extraction for film production, and, in the long run, trying to hypothesize possible applications for it, in order to follow a circular economy-based approach [48].

## 4. Conclusions

In this study, metabisulfite was used for the extraction of keratin from wool and feather waste. The method guaranteed the high solubility of the keratin with respect to other procedures; in addition, it is highly efficient and preserves the secondary structure of the protein, allowing it to identify the presence of alpha and beta structures by using Raman spectroscopy and FTIR. While detailed information at a dimensional level between 1 and 100 nm has been obtained by SAXS and corroborated by microscopy analysis of the material, confirming the optimal preservation of the intermediate structure of the refined material was conducted with respect to other extraction methods. The dimension of fibers in the liquid phase seems to fall within the range of the intermediate filament that can be harnessed for further functionalization and composition with other biopolymers. Such a study is an advantageous starting point for future implementation of devices based on the functionalized material, to be proposed in applications for biosensors, biomedical, and other added-value applications.

## Figures and Tables

**Figure 1 polymers-15-00181-f001:**
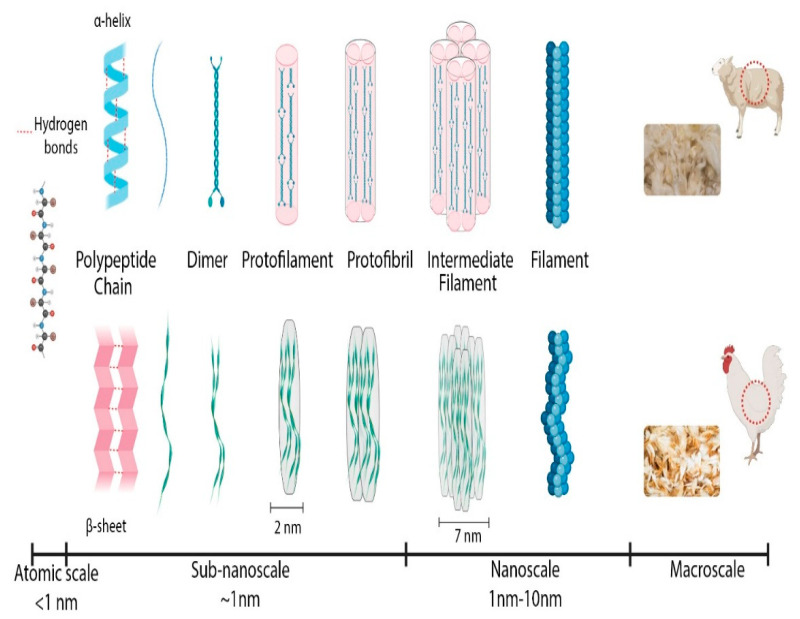
Schematic of the hierarchical structure of keratin from nanoscale to centimeter scale for wool and feathers (original drawing by S.M.).

**Figure 2 polymers-15-00181-f002:**
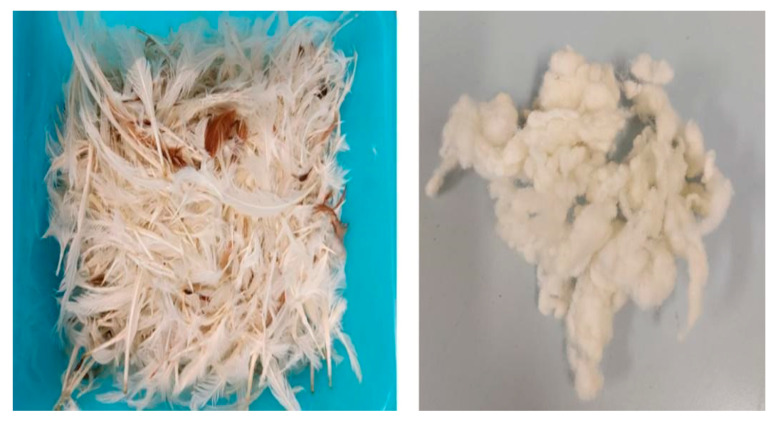
Waste as received: chicken feathers (**left**) and sheep wool (**right**).

**Figure 3 polymers-15-00181-f003:**
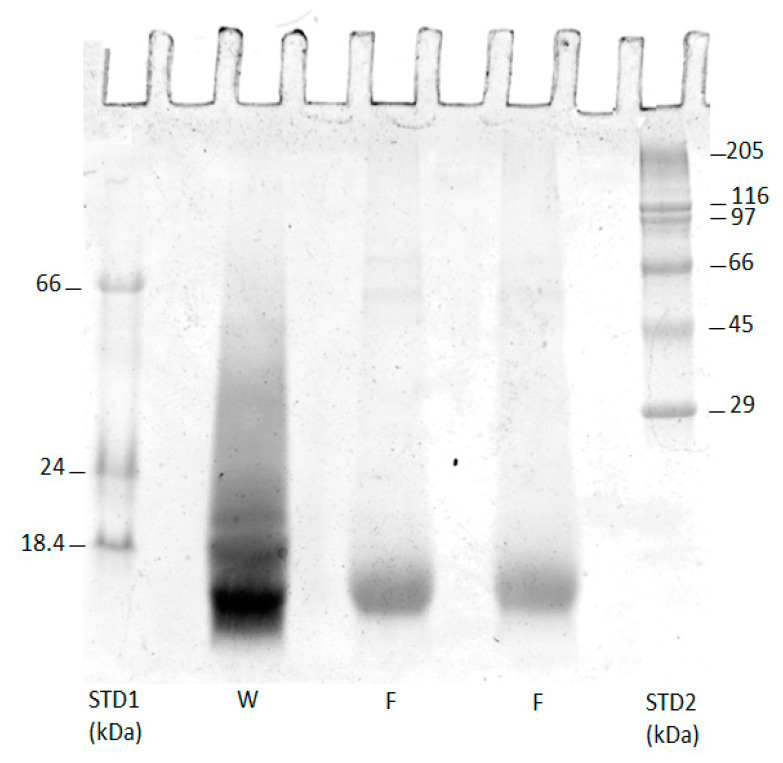
SDS-PAGE of keratin analysis. The first and last columns show the protein standard and keratin extracted from wool (W) and chicken feathers (F) using sodium metabisulfite as reducing agent.

**Figure 4 polymers-15-00181-f004:**
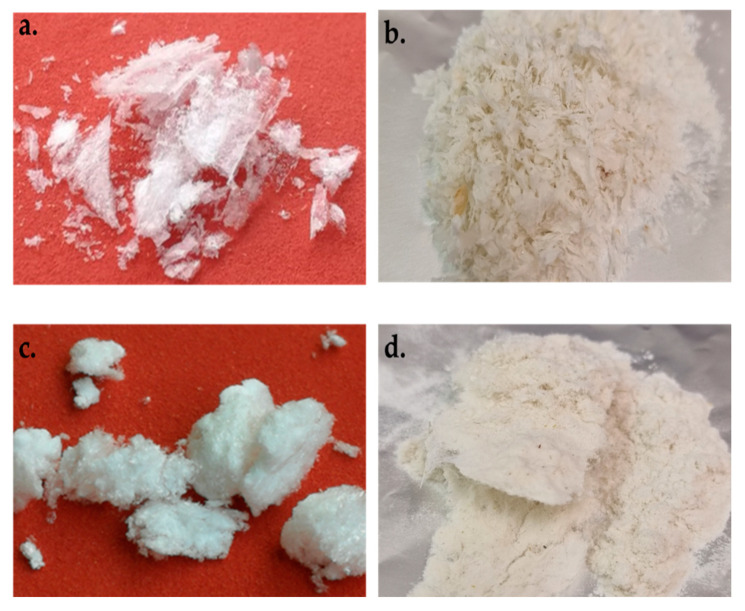
Result of keratin extraction: extracted keratin (**a**) and residue (**b**) from chicken feathers; extracted keratin (**c**) and residue (**d**) from sheep wool.

**Figure 5 polymers-15-00181-f005:**
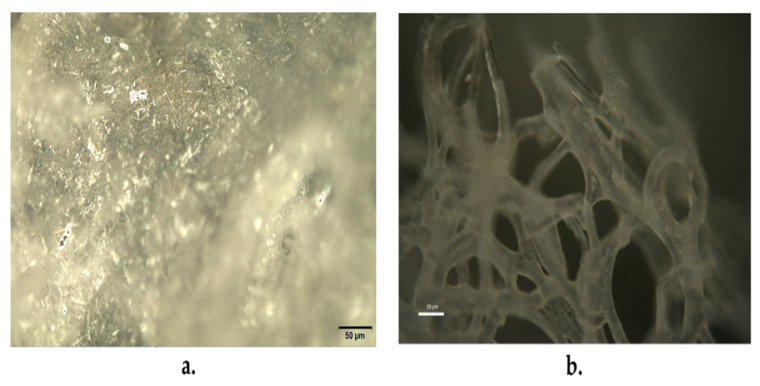
Optical micrographs of the residues of the keratin extraction process: (**a**) from chicken feathers; (**b**) from sheep wool. The scale bar indicates a 50 µm length.

**Figure 6 polymers-15-00181-f006:**
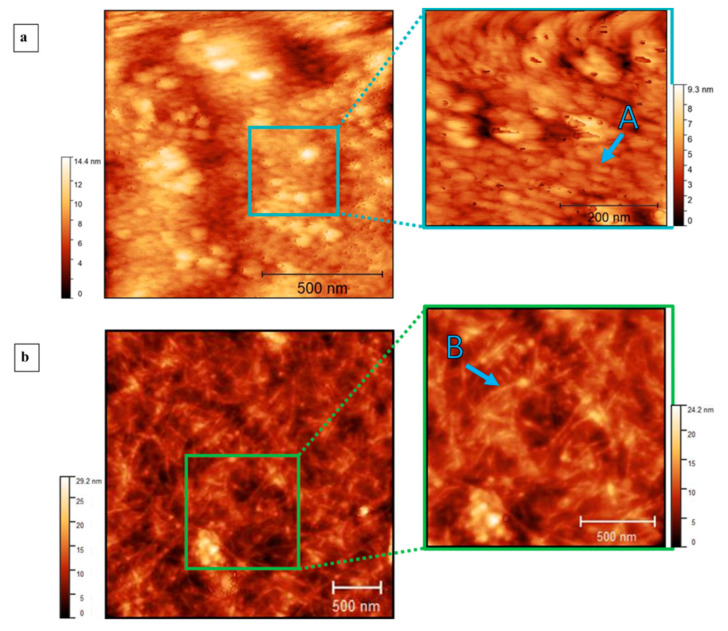
AFM images at different magnifications of keratin films obtained by drop-casting deposition, the color bar represents the variation of thickness of the materials on the substrate. (**a**) Images of keratin extracted from wool. Point A indicates the globular structure with a radius around 45 nm. (**b**) Images of keratin extracted from feathers. Point B indicates elongated structures that have lengths of approximately 500 nm.

**Figure 7 polymers-15-00181-f007:**
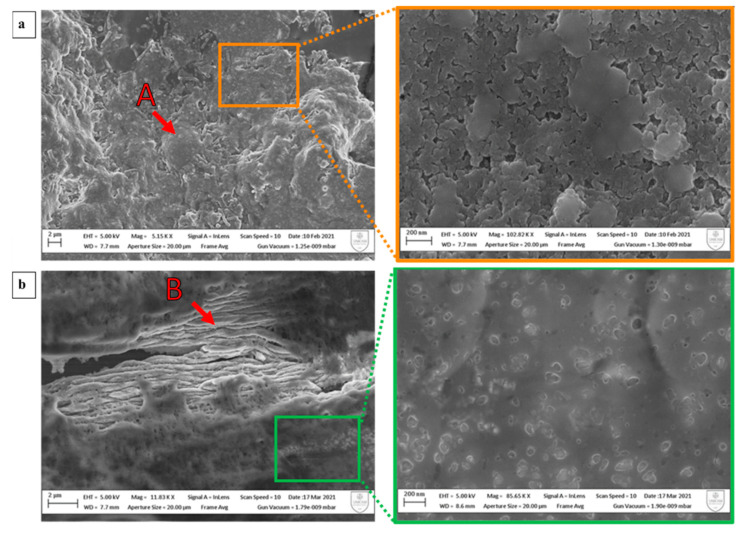
SEM images at different magnifications of keratin films obtained by drop-casting deposition. (**a**) The images of keratin extracted from wool. Point A indicates globular structure. (**b**) The images of keratin extracted from feathers. Point B indicates elongated structures, with a section of approximately 0.5 µm and a length in the order of 10 µm, hence with aspect ratio = ~20.

**Figure 8 polymers-15-00181-f008:**
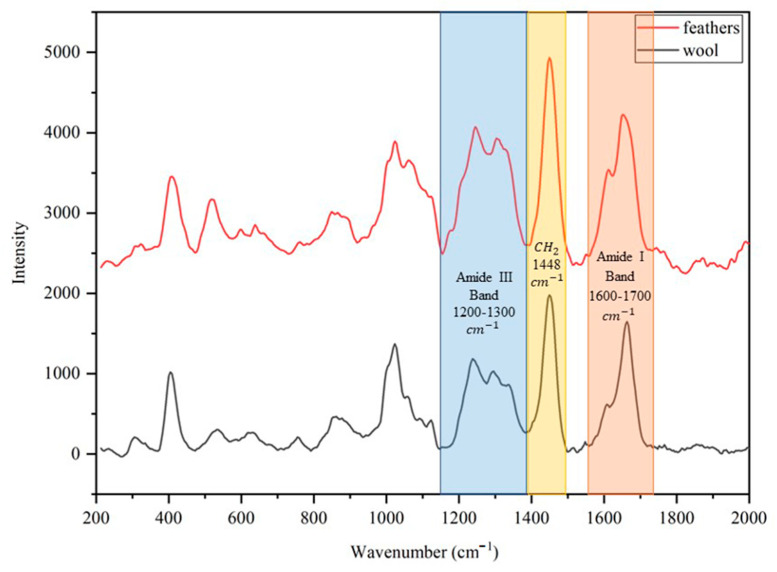
Comparison between the Raman spectrum of keratin extracted from wool (in black) and the spectrum of keratin extracted from feathers (in red).

**Figure 9 polymers-15-00181-f009:**
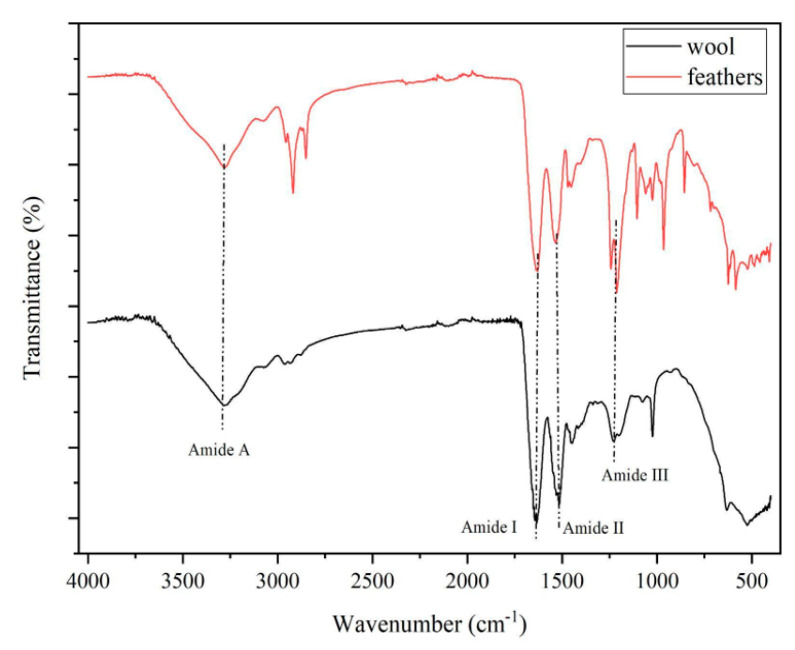
Comparison between the IR spectrum of keratin extracted from wool (in black) and the spectrum of keratin extracted from feathers (in red).

**Figure 10 polymers-15-00181-f010:**
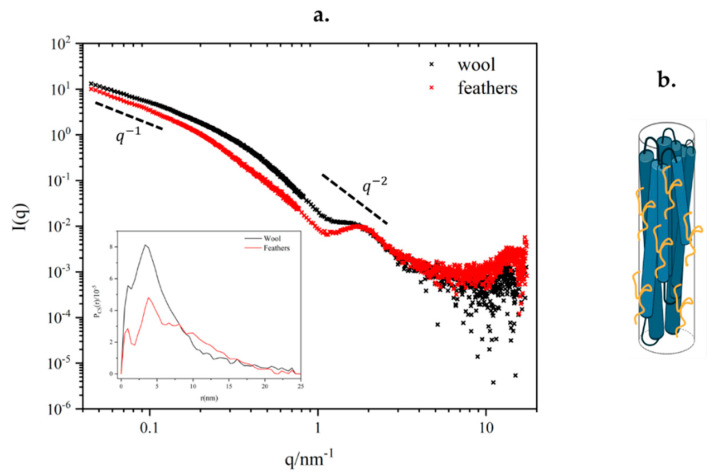
(**a**) Comparison between scattering curve of keratin extracted from wool (in black) and feathers (in red), in the inset figure, the Fourier transform of the scattering intensity; (**b**) sketch of the hypothesized fibers, which represents the possible aggregation of the smaller fibers in a bigger one; in yellow, the unfolded protein surfacing is reported.

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
