# Peer review of "Physico-Chemical Characterization of Keratin from Wool and Chicken Feathers Extracted Using Refined Chemical Methods"

_polymers, 2022, doi:10.3390/polym15010181_

Round 1
Reviewer 1 Report
The authors have done detailed background study on the extraction process of keratin. Overall, this is a novel study, needs to do some modifications for further consideration.
Specific comments:
So many linguistic errors, should check thoroughly.
industrial waste, namely sheep wool and chicken feather: Re-frame
using the most suitable purification method (metabisulfite extraction): not clear, extraction is not purification method.
structure of the protein obtained suggests: Rephrase
, estimated at 65 million tons for the former [5] and 3.1 million tons: Specify the years
There are plenty of reports available in the extraction of keratin from various sources (extracted using several methods such as chemical hydrolysis, enzymatic and microbial treatment) in literature. So, the authors should highlight the novelty of this work compared to existing data in Introduction.
The yield of the extraction method: Overall, got high yield 73 %., but what was the purity of the keratin from two sources? Why low yield (45%) for wool?
Figure 4. Optical micrographs of extracted keratin: a. From chicken feathers: Image a is not clear, cant see the structure in present image. provide a clear image.
2.3.1 Keratin film preparation Films were prepared by drop casting: This part is very important, should provide the detailed method for film preparation describing keratin concentration, solvent, incubation, film drying method, humidity, etc..
a thickness of approximately 4.8 µm: How did you measure?
Figure 5. and Figure 6. : Mark the major observations in image with symbol.
Figure 8. Comparison between the spectrum of keratin: Provide clear caption, what spectrum Raman spectroscopy or FT-IR?
4. Discussion: Overall, this section was poorly presented. Need to discuss the all the data in detail and provide the justification for the changes in results.
metabisulfite was used for the extraction of keratin : Why specifically highlighting metabisulfite, though other chemicals such as urea, EDTA, SDS, and TRIS also used in extraction process
Author Response
Please read the attached file.

Reviewer 2 Report
In this study, the wool keratin and feather keratin were extracted by the metabisulfite method, and the structural information of regenerated keratin was preliminarily characterized. A large number of experimental results are provided, but the authors need to further demonstrate the structural or functional differences of regenerated keratin. I didn't see anything in the article about self-assembly character of the keratin. In addition, the innovative nature of the paper needs to be further elucidated. I have also some comments listed below.
2.2 Extraction method
The molecular weight of feather keratin is about 10 kDa, so the author's choice of 12 kDa dialysis bag will inevitably cause the loss of keratin yield, and the dialysis efficiency is also too low, which does not have an advantage in industry.
2.3.1 Keratin film preparation
Regenerated keratin obtained by dialysis to remove reagents such as 8M urea is usually poorly soluble in water, so I don't think the regenerated keratin were able to be dissolved it in water.
3. Results
Although the author presents some structural information, the differences in keratin at the structural level are not sufficiently elucidated. It is recommended that the authors focus on comparing the structural differences between the two kinds of regenerative keratin, such as infrared and Raman spectroscopy, etc.
What about the molecular weight distribution of the two regenerative keratins?
Figure 8.
Why is feather keratin and wool keratin so different at amide III and 1000 cm-1?
What is the secondary structural composition of regenerated keratin?
Round 2
Reviewer 1 Report
The authors did not repond to the reviewer´s comments in a proper way.
For instance:
Comment no 7: but what was the purity of the keratin from two sources? Regarding this comment, no response.
Commetn no 8: Fig.4 a: The authors did not provide a clear image, instead just adjusted the brightness.
Comment no 10:How did you measure? The author respond ¨approximately 4.8 µm, as estimated from the density of the material¨. This is not a right response. How did you measure the thickness with the density?
Commetn no 13: 4. Discussion: Overall, this section was poorly presented. Need to discuss the all the data in detail and provide the justification for the changes in results: Still the discussion part is poorly described and not accetable.
and so on.....
Author Response
Comment no 7: but what was the purity of the keratin from two sources? Regarding this comment, no response.
OTHER TESTS HAVE BEEN CARRIED OUT USING ELECTROPHORESIS, WHICH CONFIRM THE MOLECULAR WEIGHT AND THE HIGH PURITY REPORTED IN LITERATURE
Comment no 8: Fig.4 a: The authors did not provide a clear image, instead just adjusted the brightness.
ANOTHER IMAGE HAS BEEN PROVIDED WITH IMPROVED CLARITY
Comment no 10: How did you measure? The author respond ¨approximately 4.8 µm, as estimated from the density of the material¨. This is not a right response. How did you measure the thickness with the density?
THE THICKNESS WAS MEASURED FROM THE WEIGHT, ASSUMING THE DENSITY AS CONSTANT THROUGHOUT THE LAYER. WE ARE AWARE THIS IS ONLY AN APPROXIMATE MEASUREMENT.
Comment no 13: 4. Discussion: Overall, this section was poorly presented. Need to discuss the all the data in detail and provide the justification for the changes in results: still the discussion part is poorly described and not acceptable.
THE DISCUSSION HAS BEEN FURTHER IMPROVED
Reviewer 2 Report
The manuscript has been performed according to previous comments. I would propose its acceptation.
Author Response
Thank you for suggesting acceptance. We are going to further improve the paper accordingly to suggestions of the other reviewer.
Round 3
Reviewer 1 Report
The authors responded well in this time. Thanks for the revision. Overall, this MS is much improved. Only some concerns below.
Comments:
Figure 1: Please provide the original image, now the image was so blurred. Letters in image were unreadable.
Figure 3. Capture the image of SDS-PAGE gel using gel documentation system, provide the image from gel-doc, now though it looks well, the gel image was not straight/aligned uniformly, so better to keep professional image. Label each well, crop the stacking gel, provide the standard marker details in method section (range of MW, source, and company) and make clear labelling in image (now the MWs of standard markers in both ends were not clear in Gel image)
The sodium dodecyl sulfate polyacrylamide gel electrophoresis (SDS-PAGE) of keratins’ preparations were: check grammar
The sodium dodecyl sulfate polyacrylamide gel electrophoresis (SDS-PAGE) of keratins’ preparations were reported in Figure 3. The molecular weights of extracted protein fractions were in the range of 12-13 kDa for feathers (F), in the range of what observed in [23], ... in addition with another less intense band around 18 kD.: Improper placement, dont present results in Method section. Move this para and Fig.3 to Results with separate subheading.
The data obtained for wool, although might appear quite low, was even slightly higher than what reported from sulfitolysis elsewhere i.e., 41% [24].: Same comment, better to move this to result and discussion section.
Figure 4.: Uneven Image size. Uniform the size of images. Image d is very undesirable format, better to replace with good one.
Figure 5.: Scale bar in both images was very small, not visible. Better to describe the size of scale bar in Fig. legend (ex. scale bar- 50 um) and remove the number 50 um from the image.
Figure 7 elongated structures with a section of about 0.5 µm: Not clear, did you mean the size of fibres indicated by point B was 0.5 µm?
Figure 10. to the right a sketch is presented: What did the sketch representing? please describe clearly.
3. Results: If you want to keep a separate Results section, then better to avoid citing reference in this section. Right now, so many citations in this section, that confuse the reader which are your results, and which are other results?
Few ex.
1200-1300 cm−1 for amide III band, 1600-1700 cm−1 for amide I band and 1448 cm−1 for CH2 group [25]: This means, these results were from the reference no 25.
Move all these citation to discussion part:
3300 cm−1 is related to amide A [26].,structure of keratin [27], N-H of amide II is visible. [28], the random coil is located around 1240-1260 cm-1 [29], due to the aliphatic CH2 vibrations of SDS [32], information on the internal structure of the fiber is obtained [33], interface according to the Porod law (q-4 ) [34].
If the authors did not want to move to discussion part, then combine the section as Results and Discussion
4. Discussion The impact on the ambient of waste from feathers.....in order to follow a circular economy-based approach [48].: Merge this section with Results section, if the authors like.
Author Response
Comments:
Figure 1: Please provide the original image, now the image was so blurred. Letters in image were unreadable.
IMAGE WAS REPLACED
Figure 3. Capture the image of SDS-PAGE gel using gel documentation system, provide the image from gel-doc, now though it looks well, the gel image was not straight/aligned uniformly, so better to keep professional image. Label each well, crop the stacking gel, provide the standard marker details in method section (range of MW, source, and company) and make clear labelling in image (now the MWs of standard markers in both ends were not clear in Gel image)
IMAGE WAS REPLACED
The sodium dodecyl sulfate polyacrylamide gel electrophoresis (SDS-PAGE) of keratins’ preparations were: check grammar
CORRECTED
The sodium dodecyl sulfate polyacrylamide gel electrophoresis (SDS-PAGE) of keratins’ preparations were reported in Figure 3. The molecular weights of extracted protein fractions were in the range of 12-13 kDa for feathers (F), in the range of what observed in [23], ... in addition with another less intense band around 18 kD.: Improper placement, dont present results in Method section. Move this para and Fig.3 to Results with separate subheading.
MOVED TO RESULTS AND DISCUSSION
The data obtained for wool, although might appear quite low, was even slightly higher than what reported from sulfitolysis elsewhere i.e., 41% [24].: Same comment, better to move this to result and discussion section.
Figure 4.: Uneven Image size. Uniform the size of images. Image d is very undesirable format, better to replace with good one.
FIGURE REPLACED
Figure 5.: Scale bar in both images was very small, not visible. Better to describe the size of scale bar in Fig. legend (ex. scale bar- 50 um) and remove the number 50 um from the image.
SCALE DESCRIBED IN THE CAPTION
Figure 7 elongated structures with a section of about 0.5 µm: Not clear, did you mean the size of fibres indicated by point B was 0.5 µm?
EXPLAINED IN THE CAPTION
Figure 10. to the right a sketch is presented: What did the sketch representing? please describe clearly.
CORRECTED
- Results: If you want to keep a separate Results section, then better to avoid citing reference in this section. Right now, so many citations in this section, that confuse the reader which are your results, and which are other results?
Few ex.
1200-1300 cm−1 for amide III band, 1600-1700 cm−1 for amide I band and 1448 cm−1 for CH2 group [25]: This means, these results were from the reference no 25.
Move all these citation to discussion part:
3300 cm−1 is related to amide A [26].,structure of keratin [27], N-H of amide II is visible. [28], the random coil is located around 1240-1260 cm-1 [29], due to the aliphatic CH2 vibrations of SDS [32], information on the internal structure of the fiber is obtained [33], interface according to the Porod law (q-4 ) [34].
If the authors did not want to move to discussion part, then combine the section as Results and Discussion
- Discussion The impact on the ambient of waste from feathers.....in order to follow a circular economy-based approach [48].: Merge this section with Results section, if the authors like.
RESULTS AND DISCUSSION SECTIONS WERE MERGED